# Potassium Application Increases Cotton (*Gossypium hirsutum* L.) Fiber Length by Improving K⁺/Na⁺ Homeostasis and Potassium Transport Capacity in the Boll-Leaf System under Moderate Salinity

Junjun Zhu [1,†], Liyuan Sun [1,†], Feiyan Ju [1], Zhuo Wang [1], Cai Xiong [1], Huilian Yu [1], Kai Yu [1], Yuyang Huo [1], Wajid Ali Khattak [1,2], Wei Hu [1], Shanshan Wang [1], Zhiguo Zhou [1] and Binglin Chen [1,*]

1 Key Laboratory of Crop Ecophysiology and Management, Ministry of Agriculture and Rural Affairs, College of Agriculture, Nanjing Agricultural University, Nanjing 210095, China
2 School of the Environment and Safety Engineering, Jiangsu University, Zhenjiang 212013, China
* Correspondence: blchen@njau.edu.cn; Tel.: +86-025-8439-6129
† These authors contributed equally to this work.

**Abstract:** Cotton has a high salt tolerance. However, due to the high salt content and low K⁺/Na⁺ ratio in saline soils, cotton yield and fiber quality are difficult to improve. To investigate the effects of potassium (K) on cotton fiber length under salt stress, a two-year bucket-based field experiment was conducted using two different cultivars (CCRI 79, salt tolerant, and Simian 3, salt sensitive). Three K rates (K0, 0 kg $K_2O$ ha⁻¹; K150, 150 kg $K_2O$ ha⁻¹; and K300, 300 kg $K_2O$ ha⁻¹) were applied at low, middle, and high soil electrical conductivities (S1, 1.7–1.8 dS m⁻¹; S2, 6.4–6.9 dS m⁻¹; and S3, 10.6–11.8 dS m⁻¹) to investigate the absorption, transport, and distribution characteristics of K⁺ and Na⁺ in the boll-leaf system (including the leaf subtending the cotton boll (LSCB), fruiting branch, boll shell, and fiber) of both cotton cultivars, as well as the relationship with fiber length. The results showed that K application (K150 and K300) significantly increased the cotton fiber length under salt stress, with the largest fiber length alleviation coefficients (AC) in the middle fruiting branches. The AC decreased with an increase in salt stress and was greater in CCRI 79 than in Simian 3. The K150 treatment (soil K⁺/Na⁺ = 1/13) completely mitigated the reduction in fiber length caused by S2 salt stress in CCRI 79, whereas the K300 treatment (soil K⁺/Na⁺ = 1/10) completely eased the reduction in fiber length caused by S2 salt stress in Simian 3. An application of K under salt stress increased the K⁺ content and K⁺/Na⁺ ratio in the soil and the organs of the boll-leaf system, regulated the K⁺/Na⁺ homeostasis in the boll-leaf system, enhanced the K⁺-selective transport coefficient ($S_{K\text{-}Na}$) in the LSCB, maintained a high K⁺/Na⁺ ratio in the fiber, and mitigated the fiber length reduction. In conclusion, the fiber length reduction in salt-tolerant cultivars was completely mitigated by K150 (i.e., soil K⁺/Na⁺ = 1/13) under moderate salt stress; however, it was not completely mitigated by K application under high salt stress.

**Keywords:** *Gossypium hirsutum* L.; fiber length; salt stress; boll-leaf system; K application; K⁺/Na⁺ homeostasis

## 1. Introduction

Soil salinization has become a major constraint to agricultural production. Of the global 14 billion hectares of cultivated land, about 1 billion hectares of land is natural saline-alkaline land. Approximately 40% of irrigated land is affected by salinization [1]. By 2050, more than 50% of the world's crop land will be saline-alkali land [2]. Saline-alkaline land in China accounts for 10.4% of the total saline lands in the world and 10.3% of the cultivated land in China, which is mainly located in the northwestern inland and coastal plains [3,4]. Salt stress has become one of the most prevalent abiotic stress limiting crop



yield and quality [5]. Cotton plants suffer from slow growth and development, yellow and withered leaves (necrosis), metabolic destruction, and even death when exposed to excessive salt stress [6–8].

Salt may adversely affect the growth, yield, and quality of cotton fibers. High salt stress leads to a significant reduction in fiber length [9,10]. The fiber quality index of salt-tolerant cultivars is higher than that of salt-sensitive cultivars [11]. $Na^+$ is the major toxic ion in salt stress [12]. With the aggravation of salt stress, the $K^+$ content declines dramatically, whereas the $Na^+$ content increases significantly [13,14]. The low $K^+/Na^+$ ratio and high $Na^+$ content in saline-alkali soils make it difficult for cotton plants to selectively absorb $K^+$, resulting in potassium (K) deficiency and ionic imbalance in cotton plants, thereby affecting the fiber quality [15,16]. Under salt stress, $Na^+$ accumulates and $K^+$ flows off, leading to an increase in the $Na^+/K^+$ ratio and the destruction of the original ion homeostasis in the plant. The plant is damaged when the $Na^+/K^+$ ratio exceeds the threshold [17]. It was documented previously that the $Na^+$ content in the aboveground organs of cotton plants changed significantly under NaCl stress [18]. As an important component of the osmotic potential of a cell, $K^+$ can be important for osmoregulation, membrane potential regulation, the co-transportation of sugars, stress adaptation, and growth [19,20]. $K^+$ homeostasis is closely related to salt tolerance [21] and is also an essential physiological mechanism for maintaining the activity and function of various enzymes in cells. It improves the salt tolerance of plants, and a higher $K^+/Na^+$ ratio is also the manifestation of ion homeostasis [10,22]. Previous studies have shown that fiber length is the most sensitive parameter to K deficiency in cotton, and a lack of $K^+$ supply significantly inhibits cotton fiber growth and ultimately the length of the cotton fiber [23]. When the $K^+$ concentration is lower than the optimal concentration, the fiber elongation decreases significantly [24]. The balance of the $K^+/Na^+$ ratio is an important physiological mechanism in plants to maintain the salt tolerance and function of various enzymes in their cells [25]. The application of K can promote the uptake of K, improve the ratio of $K^+/Na^+$ in leaves, maintain $K^+/Na^+$ homeostasis, and alleviate salt stress in plants [26,27].

The $K^+$ selective transport coefficient ($S_{K-Na}$) can represent plants' ability to transport $K^+$ to various organs under salt stress, and the higher the transport coefficient is, the stronger the ability to transport $K^+$ to various organs [28]. The photosynthetic source (the leaf subtending the cotton boll (LSCB)) and sink (cotton boll) collectively affect fiber quality [29]. The boll-leaf system, which includes the LSCB, fruiting branch, boll shell, and fiber, plays an important role in fiber growth and development. The competition and distribution of photosynthate transported to the boll is a dynamic process [30,31]. Under salt stress, the photosynthesis of cotton leaves is inhibited, which leads to incomplete and poor photosynthate transport between the boll-leaf system organs [32], resulting in shrunk cotton bolls [33]. Most previous studies have focused on the effects of salt stress on cotton leaf or boll development; however, little is known about how the boll-leaf system reconciles the effects on cotton fiber length under K treatments.

In this study, three soil salinity levels and three K rates were used to investigate the effect of K application on the cotton boll-leaf system under salt stress. The aims of this study were to investigate: (1) the alleviation effects of K application on fiber length under salt stress, (2) the relationship between cotton fiber length and $K^+$ and $Na^+$ contents and the $K^+/Na^+$ ratio in the boll-leaf system, and (3) the $K^+$ transport capacity in the boll-leaf system under salt stress.

## 2. Materials and Methods

### 2.1. Plant Material, Experimental Site, and Soil Properties

In 2019 and 2020, field experiments were conducted on coastal saline sandy loam soils at the Original Rice and Wheat Seed Farm in Dafeng, Yancheng City, Jiangsu Province (33°24′ N and 120°34′ E). The salt-tolerant cultivar CCRI 79 and the salt-sensitive variety Simian 3, with comparable medium maturity, were investigated [13]. Based on the measured soil electrical conductivity (EC), 30 cm of topsoil was removed from low-, moderate-,

and high-salt plots, and then placed into buckets containing 30 kg of soil and buried at the field's head, and a canopy was installed. Table 1 shows the soil properties before testing.

**Table 1.** Physical and chemical properties of the soils in 2019 and 2020.

| Year | Salinity Level | EC (dS m$^{-1}$) | NaCl (g kg$^{-1}$) | pH | $\rho$b (g cm$^{-3}$) | TN (g kg$^{-1}$) | Ah-N (mg kg$^{-1}$) | Av-P (mg kg$^{-1}$) | Av-K (mg kg$^{-1}$) |
|------|----------------|------------------|---------------------|-----|----------------------|------------------|----------------------|----------------------|----------------------|
| 2019 | S1 | 1.7 | 2.3 | 8.2 | 1.5 | 1.6 | 74.5 | 34.4 | 116.9 |
|      | S2 | 6.4 | 3.8 | 8.2 | 1.5 | 1.6 | 75.6 | 34.8 | 129.5 |
|      | S3 | 11.8 | 4.8 | 8.8 | 1.5 | 1.6 | 73.2 | 33.4 | 134.0 |
| 2020 | S1 | 1.8 | 1.8 | 8.1 | 1.5 | 1.7 | 79.3 | 37.0 | 160.0 |
|      | S2 | 6.9 | 3.6 | 8.1 | 1.5 | 1.7 | 77.5 | 37.5 | 136.4 |
|      | S3 | 10.6 | 4.3 | 8.4 | 1.4 | 1.7 | 82.5 | 39.0 | 170.2 |

The electrical conductivity (EC) and pH value were derived from the average value of the 5 cm soil layer before transplantation, squaring stage, blooming stage, and boll-opening stage of cotton. S1, low salinity; S2, middle salinity; S3, high salinity; $\rho$b, bulk density; TN, total nitrogen; Ah-N, alkali-hydrolysable nitrogen; Av-P, available phosphorus; Av-K, available potassium.

## 2.2. Experimental Design

During cotton growth, the low, moderate, and high soil ECs in the buckets were maintained at 1.7–1.8 dS m$^{-1}$ (S1), 6.4–6.9 dS m$^{-1}$ (S2), and 10.6–11.8 dS m$^{-1}$ (S3), respectively (Table 1). In the field, three levels of K application were established in three replications using a randomized complete block design (RCBD) under factorial arrangement: 0, 150, and 300 kg of K$_2$O ha$^{-1}$ (represented as K0, K150, and K300, respectively), corresponding to application rates of 0, 12, and 24 g of K$_2$O per bucket. Each treatment included three replicates, each with 20 cotton plants. The K fertilizer was potassium sulfate (50% K$_2$O), of which 50% was used as basal fertilizer and 50% at the first flowering stage. Urea (46% N) was used as the nitrogen (N) fertilizer, of which 30% was used as basal fertilizer, 35% at the first flowering stage, and 35% at the full flowering stage, for a total N application of 300 kg of N ha$^{-1}$. Calcium superphosphate (12% P$_2$O$_5$) was used as the phosphate (P) fertilizer, which was applied as a base fertilizer at the rate of 100 kg of P$_2$O$_5$ ha$^{-1}$. After the K application, the soil K$^+$/Na$^+$ ratio was examined (Table 2). Each year (2019 and 2020), seeds were sown on April 15, and one seedling was transplanted into each bucket at the three-leaf stage. Each bucket was irrigated to maintain a 75 $\pm$ 5% relative soil water content (RSWC) [34]. The soil moisture level of all buckets was maintained by following the method adapted by Liu et al. (2008) [35]. The cotton growth was managed according to the requirements of high-yield cropping.

**Table 2.** The soil K$^+$/Na$^+$ ratio after K application.

| Salinity Level | K Rate (kg K$_2$O ha$^{-1}$) | K$^+$/Na$^+$ of 2019 | K$^+$/Na$^+$ of 2020 | Average of K$^+$/Na$^+$ |
|----------------|------------------------------|----------------------|----------------------|--------------------------|
| S1 | 0 | 1:13 | 1:8 | 1:10 |
|    | 150 | 1:9 | 1:5 | 1:7 |
|    | 300 | 1:7 | 1:4 | 1:5 |
| S2 | 0 | 1:20 | 1:18 | 1:19 |
|    | 150 | 1:13 | 1:12 | 1:13 |
|    | 300 | 1:10 | 1:9 | 1:10 |
| S3 | 0 | 1:24 | 1:17 | 1:20 |
|    | 150 | 1:17 | 1:13 | 1:15 |
|    | 300 | 1:13 | 1:10 | 1:11 |

S1, low salinity; S2, moderate salinity; S3, high salinity.

## 2.3. Sampling and Processing

During the harvest period at the time of boll opening, all bolls with a diameter of more than 2 cm were taken from the 20 plants in each treatment. Thirty bolls were picked from the first and second nodes on the lower (2nd–4th), middle (6th–8th), and upper (10th–12th) fruiting branches of the cotton plant. White flowers were marked on the first and second

fruit nodes of the middle fruiting branches. Every 5 days from 5 to 25 days post-anthesis (DPA), 10 cotton bolls, LSCBs, and fruiting branches were selected at 9:00 a.m.

### 2.4. Cotton Fiber Length

The fiber quality of the air-dried cotton bolls was determined using the USTER HVI MF100 test system. The fiber length at 5–25 DPA was measured by the water stream method [36].

### 2.5. Concentrations of $K^+$ and $Na^+$ in the Boll-Leaf System

The middle samples of cotton plants (powdered and dried LSCB, fruiting branch, boll shell, and fiber) were digested with $H_2SO_4$-$H_2O_2$. The concentrations of $Na^+$ and $K^+$ were determined using an atomic absorption spectrophotometer (PinAAcle 900 T, America) as described by Yang et al. [37].

### 2.6. Statistical Analysis

The $K^+$-selective transport coefficient ($S_{K-Na}$) was calculated using the method of Flowers and Yeo [38] as follows:

$$S_{K-Na} = \text{sink organ } ([K^+]/[Na^+])/\text{source organ } ([K^+]/[Na^+])$$

The alleviation coefficient (AC) was calculated as follows:

$$AC_{S-KX} = (Y_{S-KX} - Y_{S-K0})/(Y_{L-K0} - Y_{S-K0}) \times 100\%$$

Here, Y denotes the fiber length, S denotes the salt level, KX denotes the application rate of K, and L denotes the low salt level.

The results were analyzed using ANOVA. The least significant difference (LSD) method was used to test the significance of the means at the $p = 0.05$ level. Variety and year were not included in the ANOVA. All calculations and statistical analyses were performed using SPPS 22.0. Figures were plotted using the OriginPro 2022 software program.

## 3. Results

### 3.1. Effect of the Application of K on Fiber Length and Its Alleviation Coefficient under Salt Stress

Salt stress significantly reduced the final fiber length of the upper, middle, and lower fruiting branches (Table 3). Under treatments S2 and S3, the fiber length of the upper fruiting branches of CCRI 79 decreased by 1.8% and 5.2% compared with S1, while those of Simian 3 decreased by 2.7% and 5.1%, respectively. The middle fruiting branches of CCRI 79 showed reductions of 1.7% and 4.4%, while those of Simian 3 showed 2.7% and 4.9% decreases in fiber length. The fiber length of the lower fruiting branches of CCRI 79 decreased by 2.2% and 4.5%, respectively, while those of Simian 3 decreased by 3.3% and 7.2%, respectively. Compared with S1, the average fiber length of CCRI 79 decreased by 1.9% and 5.1%, and for Simian 3 by 2.9% and 5.7%, under the S2 and S3 treatments, respectively. In the S2 and S3 treatments, the fiber length of the lower fruiting branches of Simian 3 decreased more than that of the upper and middle fruiting branches. In the S2 treatment, the fiber length of the lower fruiting branches of CCRI 79 declined more than that of the middle and upper fruiting branches, but in the S3 treatment, the fiber length of the upper fruiting branches decreased more than that of the middle and lower fruiting branches.

**Table 3.** The effect of K applications on fiber length and their alleviation coefficients under salt stress.

| Cultivar | Salinity Level | K Rate (kg K₂O ha⁻¹) | 2019 (mm) | | | | 2020 (mm) | | | |
|---|---|---|---|---|---|---|---|---|---|---|
| | | | Upper | Middle | Lower | Whole | Upper | Middle | Lower | Whole |
| CCRI 79 | S1 | 0 | 29.1 b | 29.2 c | 28.2 bc | 28.8 b | 29.5 b | 29.2 bc | 28.3 bc | 29.0 b |
| | | 150 | 29.9 a | 29.8 a | 28.8 ab | 29.5 a | 29.9 a | 30.0 ab | 29.0 ab | 29.6 a |
| | | 300 | 29.9 a | 30.0 a | 29.0 a | 29.6 a | 30.2 a | 30.3 a | 29.6 a | 30.0 a |
| | Mean | | 29.6 | 29.7 | 28.7 | 29.3 | 29.9 | 29.8 | 29.0 | 29.6 |
| | S2 | 0 | 28.8 bc | 28.8 d | 27.8 cd | 28.5 c | 28.9 b | 28.8 bc | 27.6 d | 28.4 c |
| | | 150 | 29.1 b | 29.4 b | 28.1 c | 28.9 b | 29.5 b | 29.3 ab | 28.5 abc | 29.1 b |
| | | 300 | 29.2 ab | 29.6 ab | 28.2 bc | 29.0 b | 29.7 a | 29.5 ab | 28.9 ab | 29.4 a |
| | Mean | | 29.0 | 29.3 | 28.0 | 28.8 | 29.4 | 29.2 | 28.3 | 29.0 |
| | S3 | 0 | 27.6 d | 27.7 f | 26.5 f | 27.3 e | 28.3 b | 28.4 c | 27.2 d | 28.0 d |
| | | 150 | 28.2 c | 28.3 e | 27.4 e | 28.0 d | 28.7 b | 28.8 ab | 27.5 c | 28.3 b |
| | | 300 | 28.4 c | 28.9 d | 27.6 d | 28.3 cd | 28.9 b | 29.0 ab | 27.7 bc | 28.5 b |
| | Mean | | 28.1 | 28.3 | 27.2 | 27.8 | 28.6 | 28.7 | 27.5 | 28.3 |
| Alleviation coefficient | S2 | 150 | 100.0 | 150.0 | 75.0 | 133.3 | 100.0 | 125.0 | 128.6 | 117.6 |
| | | 300 | 133.3 | 200.0 | 100.0 | 166.7 | 133.3 | 175.0 | 185.7 | 164.7 |
| | S3 | 150 | 40.0 | 40.0 | 41.2 | 40.0 | 33.3 | 50.0 | 27.3 | 35.5 |
| | | 300 | 53.3 | 80.0 | 47.1 | 60.0 | 50.0 | 75.0 | 45.5 | 54.8 |
| Simian 3 | S1 | 0 | 28.9 bc | 28.4 bc | 28.3 cd | 28.5 b | 29.2 bc | 28.6 cde | 28.0 ab | 28.6 de |
| | | 150 | 29.6 ab | 29.7 a | 29.3 ab | 29.5 a | 29.5 abc | 29.2 abcd | 29.4 a | 29.4 bc |
| | | 300 | 29.7 a | 29.8 a | 29.6 a | 29.7 a | 30.4 a | 29.4 a | 29.5 a | 29.8 a |
| | Mean | | 29.4 | 29.3 | 29.1 | 29.3 | 29.7 | 29.1 | 29.0 | 29.2 |
| | S2 | 0 | 28.0 d | 27.6 de | 27.7 de | 27.8 c | 28.6 c | 28.1 de | 27.4 bc | 28.0 e |
| | | 150 | 28.7 c | 28.4 bc | 28.2 c | 28.4 b | 29.0 abc | 28.6 abc | 28.1 ab | 28.6 c |
| | | 300 | 28.9 bc | 28.8 b | 28.5 bc | 28.7 b | 29.3 ab | 28.9 ab | 28.4 a | 28.9 ab |
| | Mean | | 28.5 | 28.3 | 28.1 | 28.3 | 29.0 | 28.5 | 28.0 | 28.5 |
| | S3 | 0 | 27.1 e | 27.2 e | 26.6 f | 27.0 d | 28.4 c | 27.6 e | 26.2 c | 27.4 f |
| | | 150 | 27.6 d | 27.7 cd | 27.3 e | 27.5 c | 28.6 c | 28.0 bcde | 26.7 ab | 27.8 de |
| | | 300 | 27.8 d | 28.0 cd | 27.7 de | 27.8 c | 28.7 c | 28.1 ab | 27.0 ab | 27.9 cd |
| | Mean | | 27.5 | 27.6 | 27.2 | 27.4 | 28.6 | 27.9 | 26.6 | 27.7 |
| Alleviation coefficient | S2 | 150 | 77.8 | 100.0 | 83.3 | 87.0 | 66.7 | 100.0 | 116.7 | 94.1 |
| | | 300 | 100.0 | 150.0 | 133.3 | 126.1 | 166.7 | 160.0 | 166.7 | 147.1 |
| | S3 | 150 | 27.8 | 41.7 | 41.2 | 36.2 | 25.0 | 40.0 | 27.8 | 30.6 |
| | | 300 | 38.9 | 66.7 | 64.7 | 55.3 | 37.5 | 50.0 | 44.4 | 44.4 |
| | Salinity(S) | | | | | ** | | | | ** |
| | K rate(K) | | | | | ** | | | | ** |
| | Cultivar(C) | | | | | ** | | | | ** |
| | S × K | | | | | ** | | | | ** |
| | S × C | | | | | ** | | | | ** |
| | K × C | | | | | ** | | | | ** |

Values followed by different letters within the same column are significantly different at the $p = 0.05$ probability level. Each value represents the mean of three replications. ** indicates significant differences at the 0.01 probability levels; $n = 18$, $R_{0.05} = 0.468$, and $R_{0.01} = 0.590$. S1, low salinity; S2, moderate salinity; S3, high salinity; Upper, 10th–12th fruiting branches; Middle, 6th–8th fruiting branches; Lower, 2nd–4th fruiting branches; Whole, whole-plant fruiting branches.

At the same salt level, different K application rates were effective at alleviating the adverse effects of salt stress on fiber length. The alleviation effect of K applications gradually decreased as the salinity level increased. Over the years with the S2 salt level, the average alleviation coefficients (ACs) of the upper fruiting branches of CCRI 79 under the treatments K150 and K300 were 100.0% and 133.3%, respectively, compared with K0, and those for Simian 3 were 72.3% and 133.4%, respectively. The ACs of the middle fruiting branches of CCRI 79 were 137.5% and 187.5% for the treatments K150 and K300, respectively, and 100.0% and 155.0% for Simian 3. The whole-plant ACs of the lower fruiting branches of CCRI 79 were 101.8% and 142.9%, respectively, while those of Simian 3 were 100.0% and 150.0%. The ACs of CCRI 79 were 125.5% and 165.7% for the K150 and K300 treatments, respectively, and those of Simian 3 were 90.6% and 136.6%. Under the S3 salt level, the ACs of the upper fruiting branches of CCRI 79 under the K150 and K300 treatments were 36.7% and 51.7%, respectively, compared with K0, while those of Simian 3 were 26.4% and 38.2%. The ACs of the middle fruiting branches of CCRI 79 were 45.0% and 77.5%, while those of Simian 3 were 40.9% and 58.4%. The ACs of the lower fruiting branches of CCRI 79 were 34.3% and 46.3%, and those of Simian 3 were 34.5% and 54.6%. Under the K150 and K300 treatments, the whole-plant fiber lengths of CCRI 79 were 37.8% and 57.4%, respectively, while those for Simian 3 were 33.4% and 49.9%. The alleviating effect of K application gradually became weaker as the salinity level increased. At the S2 and S3 salt levels, the ACs of the middle fruiting branches of the two varieties were higher than those in the

lower and upper parts, and those of CCRI 79 were higher than those of Simian 3. During fiber elongation, the cotton fiber length increased rapidly before 15 DPA and reached its maximum elongation rate at 10–15 DPA. After 20 DPA, the growth of the cotton fiber length slowed down and gradually stabilized. The time point of 15 DPA can be considered a key turning point in the study of fiber length. Salt stress significantly reduced the fiber length of the two varieties, while K applications effectively increased the fiber length under salt stress (Figure 1). The results showed that, at 15 DPA, the cotton fiber length of CCRI 79 decreased by 7.0% and 13.4% under the S2 and S3 treatments, respectively, compared with S1, while that of Simian 3 decreased by 2.8% and 9.8% for the two treatments. At the S2 salt level, the ACs of CCRI 79 were 76.4% and 135.7% for the K150 and K300 treatments, while those of Simian 3 were 68.9% and 117.3%, respectively. At the S3 salt level, the ACs of CCRI 79 were 29.7% and 59.5%, while those of Simian 3 were 25.0% and 55.1%. Salt stress shortened the fiber length and different K application rates effectively alleviated its adverse effects.

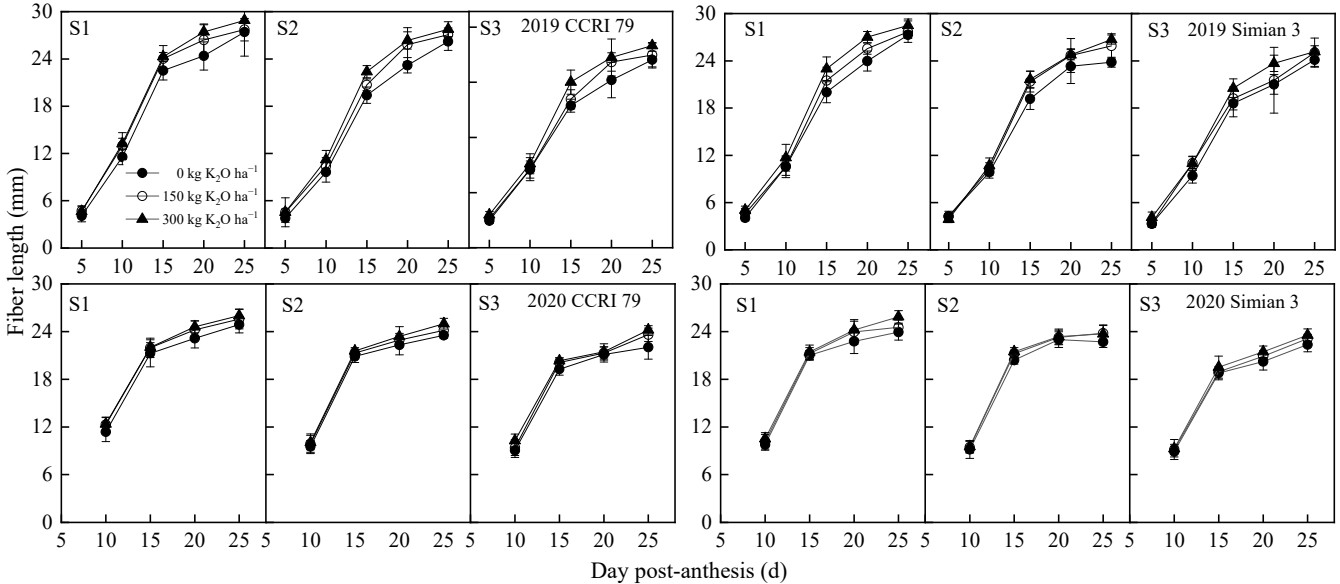

**Figure 1.** Effect of K application on dynamic changes in cotton fiber length from 5 to 25 DPA under salt stress. S1, low salinity; S2, moderate salinity; S3, high salinity.

### 3.2. Effect of K Application on the K$^+$ Content in Various Organs of the Boll-Leaf System under Salt Stress

The results showed that the amount of K$^+$ in the LSCB, fruiting branch, boll shell, and fiber of the boll-leaf system decreased as the salt stress worsened at 15 DPA, and that K applications increased the content of K$^+$ in each organ (Figure 2). The K$^+$ content in each organ of CCRI 79 was generally higher than that of Simian 3.

At the S1 salt level, compared with K0, the K$^+$ content in the LSCB of CCRI 79 increased by 14.9% and 31.2% for the K150 and K300 treatments, respectively, while those of the LSCB of Simian 3 increased by 10.6% and 18.8%, respectively. At the S2 salt level, the K$^+$ content of CCRI 79 increased by 4.4% and 11.6%, respectively, while that of Simian 3 increased by 5.4% and 13.7%. At the S3 salt level, the K$^+$ content of CCRI 79 increased by 7.6% and 12.2%, respectively, while that of Simian 3 increased by 7.5% and 15.3%.

At the S1 salt level, the K$^+$ content of CCRI 79 in fruiting branches increased by 4.0% and 13.3% for K150 and K300 compared with K0, while that of Simian 3 increased by 4.9% and 13.7%, respectively. At the S2 salt level, the K$^+$ content of CCRI 79 fruiting branches increased by 5.2% and 13.8%, while that of Simian 3 increased by 16.2% and 20.9%, respectively. At the S3 salt level, the K$^+$ content in fruiting branches of CCRI 79 increased by 5.6% and 12.3%, while that of Simian 3 rose by 11.8% and 18.2%, respectively.

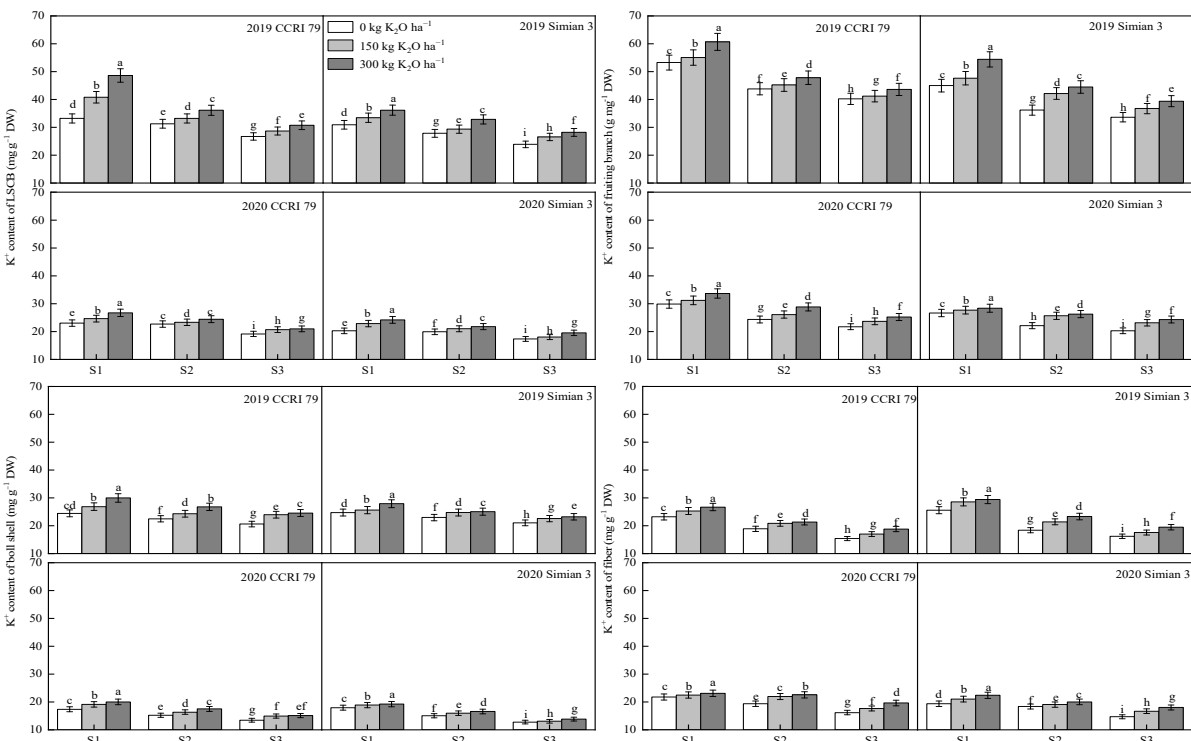

**Figure 2.** Effect of K application on the content of $K^+$ in the boll-leaf system at 15 DPA under salt stress. S1, low salinity; S2, moderate salinity; S3, high salinity. Different letters denote significant differences between the mean values ($p < 0.05$).

At the S1 salt level, the $K^+$ content of the boll shell of CCRI 79 increased by 9.9% and 18.9% in the K150 and K300 treatments, respectively, and by 4.4% and 10.0% for Simian 3. At the S2 salt level, the content of CCRI 79 increased by 7.8% and 17.1%, while for Simian 3 it increased by 7.0% and 9.6%, respectively. At the S3 salt level, the $K^+$ content of CCRI 79 increased by 13.8% and 16.0%, respectively, while that of Simian 3 increased by 4.8% and 9.5%. Under salt stress, the $K^+$ content of the boll shell of CCRI 79 decreased less than that of Simian 3, and the increase in the $K^+$ content after the K application was higher than that of Simian 3.

At the S1 salt level, the fiber $K^+$ content of CCRI 79 increased by 6.1% and 10.6% in the K150 and K300 treatments, respectively, while it increased by 1.1% and 15.3% of Simian 3, respectively. At the S2 salt level, the $K^+$ content of CCRI 79 increased by 12.0% and 14.8%, respectively, while that of Simian 3 increased by 10.1% and 18.0%, respectively. At the S3 salt level, the $K^+$ content of CCRI 79 increased by 13.8% and 16.0%, respectively, and that of Simian 3 increased by 9.9% and 21.9%.

### 3.3. Effect of K Application on $Na^+$ Content in Boll-Leaf System Organs under Salt Stress

With increasing salt stress, the $Na^+$ content of the LSCB, fruiting branch, boll shell, and fiber of the boll-leaf system increased, whereas K applications decreased the $Na^+$ content of each organ at 15 DPA (Figure 3). The $Na^+$ content in each organ of CCRI 79 was mostly lower than that of Simian 3.

At the S1 salt level, the $Na^+$ content in the LSCB of CCRI 79 decreased by 14.5% and 33.2% compared to K0 under the K150 and K300 treatments, respectively, while that of Simian 3 decreased by 12.2% and 22.8%. At the S2 salt level, the $Na^+$ content of CCRI 79 was reduced by 10.8% and 25.9%, and that of Simian 3 was reduced by 31.7% and 16.6%, respectively. At the S3 salt level, the $Na^+$ content of CCRI 79 decreased by 9.5% and 22.3%, while that of Simian 3 decreased by 12.4% and 21.1%, respectively. The $Na^+$ reduction rate in the LSCB of CCRI 79 was higher than that of Simian 3 under the K300 treatment.

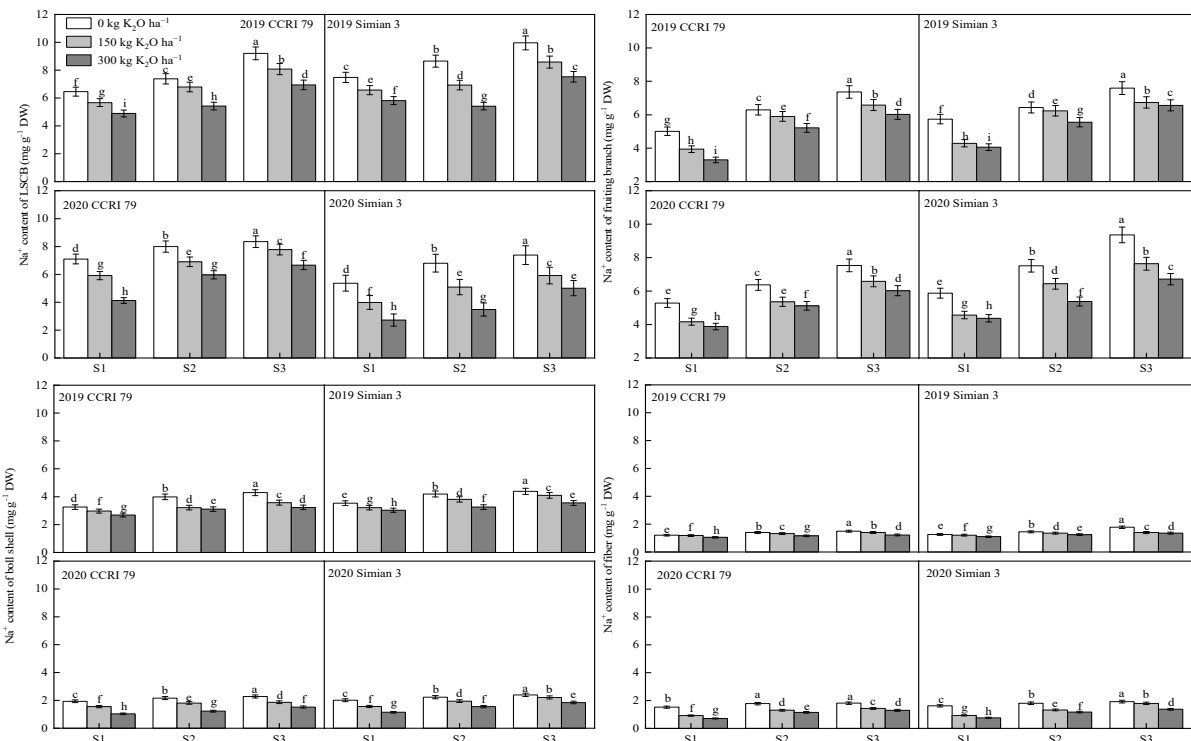

**Figure 3.** Effect of K application on the content of Na$^+$ in the boll-leaf system at 15 DPA under salt stress. S1, low salinity; S2, moderate salinity; S3, high salinity. Different letters denote significant differences between the mean values($p < 0.05$).

At the S1 salt level, the Na$^+$ content of CCRI 79 fruiting branches decreased by 21.3% and 30.4% compared with K0 under the K150 and K300 treatments, respectively, while it decreased by 23.7% and 27.5% for Simian 3. At the S2 salt level, the Na$^+$ content of CCRI 79 fruiting branches decreased by 11.0% and 18.4%, respectively, while it decreased by 8.7% and 21.0% for Simian 3. At the S3 salt level, the Na$^+$ content of the CCRI 79 fruiting branches decreased by 11.6% and 19.1%, respectively, while that of Simian 3 decreased by 14.9% and 20.9%, respectively.

At the S1 salt level, the Na$^+$ content of the CCRI 79 boll shell decreased by 14.3% and 32.0% under the K150 and K300 treatments, respectively, while that of Simian 3 decreased by 15.8% and 28.6%. At the S2 salt level, the Na$^+$ content of the CCRI 79 boll shell decreased by 17.7% and 32.5%, respectively, while that of Simian 3 decreased by 10.9% and 26.3%, respectively. At the S3 salt level, the Na$^+$ content of the CCRI 79 boll shell decreased by 17.4% and 28.9%, respectively, while that of Simian 3 decreased by 7.2% and 21.0%, respectively. The Na$^+$ reduction rate of the CCRI 79 boll shell was higher than that of Simian 3 under the K300 treatment.

At the S1 salt level, the Na$^+$ content of the fiber of CCRI 79 decreased by 20.8% and 32.8% for the K150 and K300 treatments, respectively, while that of Simian 3 decreased by 23.2% and 32.8%, respectively. At the S2 salt level, the Na$^+$ content of CCRI 79 decreased by 16.0% and 26.2%, while that of Simian 3 decreased by 16.3% and 24.8%, respectively. At the S3 salt level, the content of CCRI 79 decreased by 13.6% and 23.2%, while that of Simian 3 decreased by 13.9% and 26.1%, respectively.

### 3.4. Effect of K Application on the K$^+$/Na$^+$ Ratio in the Boll-Leaf System under Salt Stress

At 15 DPA, the K$^+$/Na$^+$ ratio in the LSCB, fruiting branch, boll shell, and fiber of the boll-leaf system declined continuously with increasing salt stress (Figure 4). The effect of K300 was greater than that of K150. The ratio of K$^+$/Na$^+$ in the cotton fibers was higher than that in other organs. CCRI 79 had a greater K$^+$/Na$^+$ ratio than Simian 3.

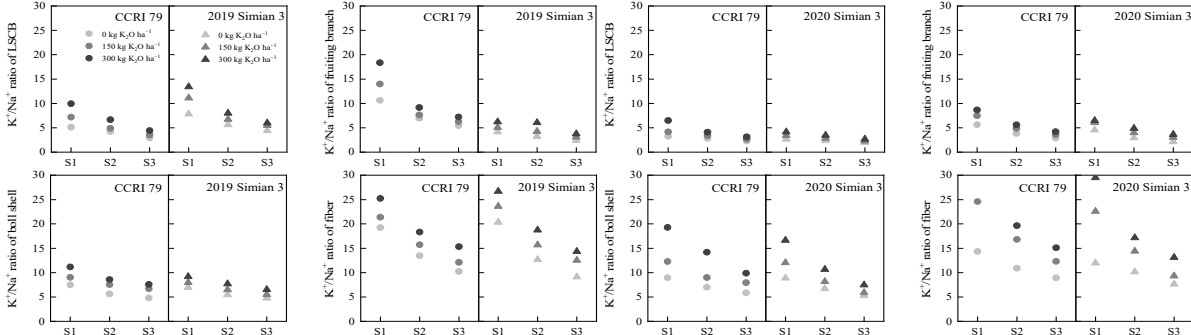

**Figure 4.** Effect of K application on the $K^+/Na^+$ ratio in the boll-leaf system at 15 DPA under salt stress. S1, low salinity level; S2, moderate salinity level; S3, high salinity level.

In conclusion, salt stress significantly reduced the $K^+$ content, increased the $Na^+$ content, and decreased the $K^+/Na^+$ ratio in the boll-leaf system of both cultivars. The $K^+$ content of each organ in the boll-leaf system tended to decrease in the order of fruiting branch, LSCB, boll shell, and fiber, while the fiber maintained a higher $K^+/Na^+$ ratio.

*3.5. Correlations of Cotton Fiber Length with $K^+$ and $Na^+$ Contents in the Boll-Leaf System*

The correlation analysis between the cotton fiber length, the $K^+$ and $Na^+$ contents, and the $K^+/Na^+$ ratio in the LSCB, fruiting branch, boll shell, and fiber at 15 DPA can be seen in Table 4. A significant positive correlation ($p < 0.01$) was observed between the fiber length, the $K^+$ content, and the $K^+/Na^+$ ratio in various organs. However, there was a significant negative correlation ($p < 0.01$) between the fiber length and the $Na^+$ content. The correlation coefficient of fiber length with the $K^+$ content of the fiber was higher than that in other organs.

**Table 4.** Correlation of cotton fiber length (mm) with $K^+$ and $Na^+$ content.

| Year | Parameter | LSCB | Fruiting Branch | Boll Shell | Fiber |
|------|-----------|------|-----------------|------------|-------|
| | $K^+$ | 0.8820 ** | 0.8943 ** | 0.9076 ** | 0.9726 ** |
| 2019 | $Na^+$ | −0.9138 ** | −0.9195 ** | −0.8896 ** | −0.8414 ** |
| | $K^+/Na^+$ | 0.9089 ** | 0.8895 ** | 0.9221 ** | 0.8599 ** |
| | $K^+$ | 0.9239 ** | 0.8666 ** | 0.9226 ** | 0.9436 ** |
| 2020 | $Na^+$ | −0.7864 ** | −0.9183 ** | −0.7408 ** | −0.7512 ** |
| | $K^+/Na^+$ | 0.7776 ** | 0.8666 ** | 0.7752 ** | 0.7718 ** |

** indicates significant differences at the 0.01 probability level.

*3.6. The K Transport Capacity in the Boll-Leaf System under Salt Stress*

The $K^+$-selective transport coefficient ($S_{K-Na}$) from the LSCB to the fiber of the two varieties decreased consistently as the salt stress increased at 15 DPA (Figure 5). The $S_{K-Na}$ from the LSCB to the fiber of CCRI 79 decreased less with increasing salt stress, with S2 and S3 decreasing by 6.2% and 0.3%, respectively, compared with S1, while Simian 3 decreased more, with a decrease in S2 and S3 of 21.5% and 22.2%, respectively.

Under salt stress, the effect of K application increased the $S_{K-Na}$ from the LSCB to the fiber, but the effect of the K application gradually decreased with the decline in salt stress. At the S1 salt level, the $S_{K-Na}$ levels of CCRI 79 from the LSCB to the fiber increased by 15.5% and 40.5%, in the K150 and K300 treatments, respectively, while those of Simian 3 increased by 27.5% and 36.4% for the above treatments compared with K0. At the S2 salt level, compared with K0, the $S_{K-Na}$ values of CCRI 79 from the LSCB to the fiber increased by 20.3% and 23.5%, respectively, while those of Simian 3 increased by 17.0% and 21.8%. At the S3 salt level, the $S_{K-Na}$ values of CCRI 79 increased by 10.2% and 13.6%, respectively, while those of Simian 3 increased by 2.6% and 16.4%, respectively.

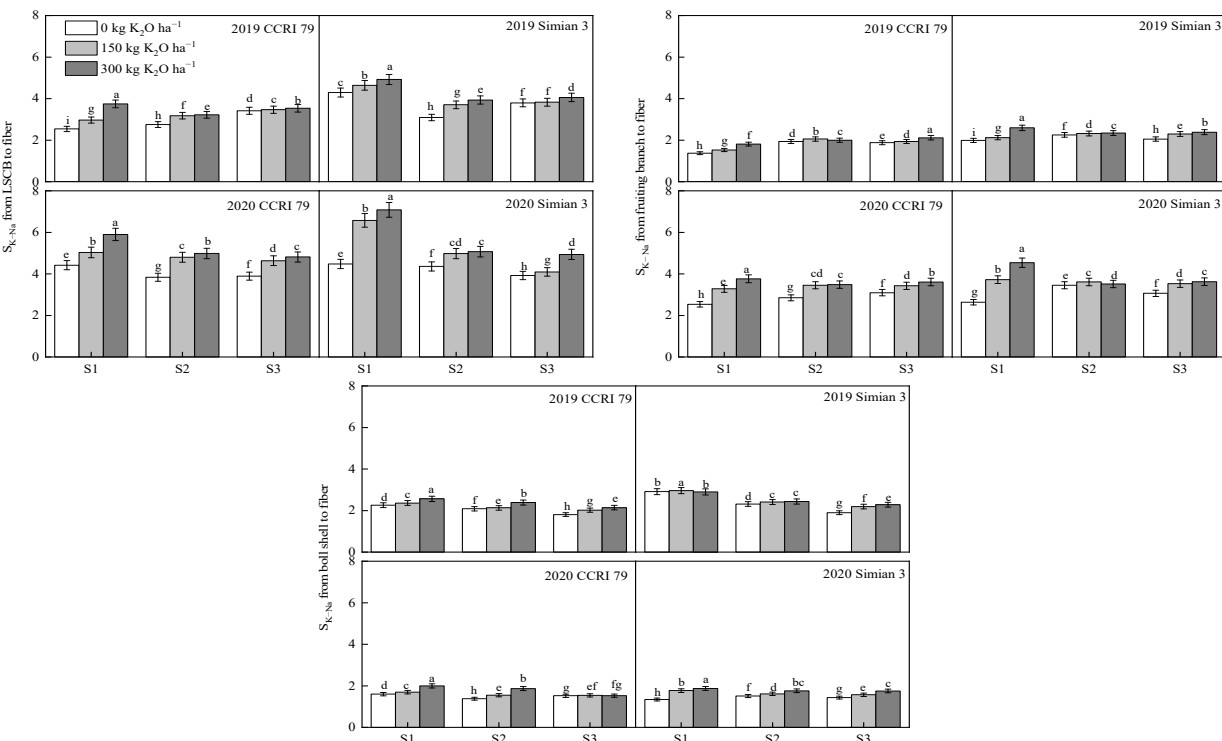

**Figure 5.** The effect of K application on $S_{KNa}$ at 15 DPA under salt stress. S1, low salinity level; S2, moderate salinity level; S3, high salinity level. Different letters denote significant differences between the mean values ($p < 0.05$).

With increasing salt stress, the $S_{K-Na}$ value from the fruiting branch to the fiber of CCRI 79 increased at 15 DPA, and the values for S2 and S3 increased by 14.6% and 15.8%, respectively, compared with that of S1. The values for Simian 3 increased by 0.1% under the S2 salt level and decreased by 2.9% under the S3 salt level. The K application increased the $S_{K-Na}$ from the fruiting branch to the fiber in both varieties under salt stress. At the S1 salt level, the $S_{K-Na}$ from the fruiting branch to the fiber increased by 20.4% and 40.3% in CCRI 79 compared with K0 under the K150 and K300 treatments, respectively, while it increased by 24.1% and 51.3% for Simian 3. At the S2 salt level, the $S_{K-Na}$ values from the fruiting branch to the fiber of CCRI 79 increased by 13.7% and 12.8%, respectively, while those of Simian 3 increased by 3.8% and 2.8%, respectively. At the S3 salt level, the $S_{K-Na}$ values from the fruiting branch to the fiber of CCRI 79 increased by 6.9% and 14.6%, respectively, while those of Simian 3 increased by 13.5% and 17.3%, respectively.

The $S_{K-Na}$ value from the boll shell to the fiber of both varieties decreased with increasing stress at 15 DPA. Under S2 and S3, the $S_{K-Na}$ from the boll shell to the fiber decreased by 8.7% and 15.1% for CCRI 79, respectively, compared with S1, and by 10.2% and 16.0% for Simian 3, respectively. At the S1 salt level, the $S_{K-Na}$ values from the boll shell to the fiber of CCRI 79 increased by 5.3% and 19.3% compared with K0 at K150 and K300, respectively, while those of Simian 3 increased by 16.7% and 19.5%, respectively. At the S2 salt level, the $S_{K-Na}$ values from the boll shell to the fiber of CCRI 79 increased by 7.1% and 24.9%, respectively, while those of Simian 3 increased by 5.5% and 11.0%, respectively. At the S3 salt level, the $S_{K-Na}$ values from the boll shell to the fiber of CCRI 79 increased by 6.6% and 9.3%, respectively, while those of Simian 3 increased by 12.5% and 21.2%, respectively.

With an increase in salinity, the $S_{K-Na}$ from the LSCB, fruiting branches, and boll shells of the boll-leaf system to the fiber changed, with the $S_{K-Na}$ from the fruiting branches to the fiber increasing while the $S_{K-Na}$ from the LSCB to the fiber and the $S_{K-Na}$ from the boll shells to the fiber decreased.

## 4. Discussion

### 4.1. Alleviation Effects of K Application on Cotton Fiber Length under Salt Stress

The final length of the cotton fiber decreased with increasing salt stress, which was consistent with the results of previous studies [11,39]. Plant salt tolerance can be improved by increasing the endogenous K levels with K fertilizer. The alleviating effect of K fertilizer on cotton growth and yield under saline conditions has also been demonstrated [10,40]. With the aggravation of salt stress, the alleviating effect of K fertilizer on fiber length decreased significantly (Table 3). High salt stress is still the key factor limiting fiber length [15,41]. In this study, we discovered that the ACs of whole-plant fiber length under moderate salt stress (S2) were 117.6–133.3% and 164.7–166.7% for the salt-tolerant variety (CCRI 79) under the K150 and K300 treatments, and 87.0–94.1% (AC < 100%) and 126.1–147.1% for the salt-sensitive variety (Simian 3), respectively. The findings indicate that salt-tolerant varieties under S2 salt stress were able to mitigate the decrease in fiber length under lower K applications (K150) and salt-sensitive varieties were able to mitigate the decrease under higher K applications (K300). Under high salt stress (S3), the ACs under K300 were 54.8–60.0% (AC < 100%) for CCRI 79 and 44.4–55.3% (AC < 100%) for Simian 3, indicating that the K application could no longer alleviate the decrease in fiber length caused by S3 salt stress. K applications attenuated the decrease in fiber length, and K300 had a better alleviating effect. Under salt stress, the fiber length decreased more in the lower fruiting branches, while the K application primarily increased the AC in the middle fruiting branches.

### 4.2. Effects of K Application on $K^+$ and $Na^+$ Contents and the $K^+/Na^+$ Ratio in the Boll-Leaf System under Salt Stress

Under salt stress, $Na^+$, as the main toxic ion, accumulates in large quantities [12], limiting crop-selective $K^+$ uptake, which results in K shortages in cotton plants and an imbalance in the $K^+/Na^+$ ratio [16]. Previous studies have shown that cotton selectively absorbs $Na^+$ and $K^+$ ions and maintains a high $K^+/Na^+$ ratio, both of which are essential manifestations of salt tolerance in plants [42,43]. Salt-tolerant crops can maintain low $Na^+$ accumulation at growth points because of the different capacities of $Na^+$ transport and accumulation in different tissues or organs [44–46]. Similarly, our data indicated that salt stress significantly reduced the $K^+$ content, increased the $Na^+$ content, and decreased the $K^+/Na^+$ ratio in the boll-leaf system of both the studied cultivars (Figures 2–4). The $K^+$ content and $K^+/Na^+$ ratio in the organs of the salt-tolerant cultivar CCRI 79 were higher than those of the salt-sensitive cultivar Simian 3, and the trend in $Na^+$ content was reversed.

K applications improved the ion homeostasis, photosynthetic capacity, and carbohydrate metabolism of LSCBs [47]. A K application to saline soils increased the $K^+$ content and $K^+/Na^+$ ratio in the soil (Table 2). Similarly, in the organs of the boll-leaf system, $K^+$ was also decreased, but $Na^+$ was lower, with K300 having a larger effect than K150 (Figures 2–4). The $K^+$ content of each organ in the boll-leaf system tended to decrease in the order of fruiting branch, LSCB, boll shell, and fiber, while the fiber was able to maintain a higher $K^+/Na^+$ ratio, which was beneficial for improving fiber quality.

Salt stress has been shown to exacerbate plant metabolism via a $K^+/Na^+$ imbalance [2], and K applications can mitigate $Na^+$ toxicity [14,48]. An adequate K supply can prevent the cotton fiber shortening caused by salt stress. The boll-leaf system is an example of the critical source-sink interaction. The boll shell contributes to seed and fiber growth and development by providing nutrition, support, and protection [49], and also by influencing fiber length. A correlation analysis revealed that fiber length was considerably positively correlated with the $K^+$ content and the $K^+/Na^+$ ratio in each organ and significantly negatively correlated with the $Na^+$ content, and the correlation coefficient between the fiber length and the $K^+$ content was higher in the fiber than in the other organs (Table 4).

*4.3. Effect of K Application on $K^+$ Transport Capacity in the Boll-Leaf System under Salt Stress*

Ion-selective transport coefficients can represent, to some extent, a plant's ability to transport ions to various organs. The higher the ion-selective transport coefficient under salt stress is, the better the ability to transfer ions to different organs [29,50,51]. In this study, as the salt stress increased, the $S_{K-Na}$ from the LSCB, fruiting branches, and boll shells of the boll-leaf system to the fiber changed. The $S_{K-Na}$ from the fruiting branches to the fiber increased, while the $S_{K-Na}$ from the LSCB to the fiber and the $S_{K-Na}$ from the boll shells to the fiber decreased (Figure 5). These findings are similar to those found in the roots and leaves of tree species under salt stress [52].

Under salt stress, the addition of K can increase the $K^+$ content and $S_{K-Na}$. To maintain $K^+/Na^+$ homeostasis, the capacity of $K^+$ to be selectively absorbed is increased during the mild phase of salt stress. The effect is stressful and mild. Aggressive stress causes plants to store more $Na^+$, leading to an ion imbalance until $K^+$ can no longer be absorbed. Under high salt stress, the capacity of the boll-leaf system to selectively transport $K^+$ decreases, possibly because of an increase in $Na^+$, leading to smaller stomatal openings in plant leaves, less transpiration, a greater cell membrane permeability, and ionic toxicity [52]. Furthermore, the effect of K300 was greater than that of K150, and the $S_{K-Na}$ connection from the LSCB to the fiber was stronger than that from the fruiting branch to the fiber or from the boll shell to the fiber. As the primary source organ, the LSCB may continue $K^+$ transport and maintain metabolism and growth.

## 5. Conclusions

Overall, the experimental results showed that under salt stress in 2019 and 2020, (1) K applications increased the AC of the fiber length of the middle fruiting branches more than that of the lower and upper branches. K150 (i.e., $K^+/Na^+$ = 1/13) completely alleviated the S2 salt stress-induced reduction in fiber length of the salt-tolerant variety (CCRI 79), while K300 (i.e., $K^+/Na^+$ = 1/10) entirely alleviated the reduction in fiber length of the salt-sensitive variety (Simian 3) caused by S2 salt stress. (2) A K application under salt stress increased the $K^+$ content and $K^+/Na^+$ ratio, decreased the $Na^+$ content in saline soils and the organs of the boll-leaf system, regulated $K^+/Na^+$ homeostasis in the boll-leaf system, maintained a high $K^+/Na^+$ ratio in the fiber, and alleviated the decrease in fiber length caused by salt stress. (3) The $S_{K-Na}$ in the boll-leaf system decreased with the aggressiveness of salt stress. A K application increased the $K^+$ content and $S_{K-Na}$. The $S_{K-Na}$ from the LSCB to the fiber was higher, which promoted fiber elongation. As a result, the regulatory effects of increasing K fertilizer application rates on the $K^+/Na^+$ homeostasis and the potassium transport capacity in the boll-leaf system under salt-stress conditions are revealed, laying the theoretical foundation for increasing cotton fiber length under salt stress.

In the future, the absorption, transport, and distribution characteristics of $K^+$ and $Na^+$ in the roots of different salt-tolerant cotton cultivars will be further discussed based on our present studies.

**Author Contributions:** Conceptualization, J.Z., L.S., Z.Z. and B.C.; data collection, J.Z., L.S., Y.H. and K.Y.; methodology, F.J., H.Y. and W.H.; formal analysis, Z.W., C.X. and S.W.; writing—original draft, J.Z. and L.S.; writing—review and editing, J.Z., L.S., F.J., W.A.K., W.H. and B.C.; funding acquisition, B.C. and Z.Z. All authors have read and agreed to the published version of the manuscript.

**Funding:** This research was funded by the Jiangsu Agriculture Science and Technology Innovation Fund (CX(22)2015); the National Natural Science Foundation of China (31671623); the Jiangsu Postdoctoral Research Funding Program (2021K542C); and the Earmarked Fund for China Agriculture Research System (CARS-15-14).

**Data Availability Statement:** The data presented in this study are available within the article.

**Acknowledgments:** This work was supported by the Collaborative Innovation Center for Modern Crop Production co-sponsored by Province and Ministry, Nanjing, China, with the help of agronomists from the Dafeng Original Seed Farmin Yancheng, Jiangsu Province of China.

**Conflicts of Interest:** The authors declare no conflict of interest.

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
