# Peer review of "Potassium Application Increases Cotton (Gossypium hirsutum L.) Fiber Length by Improving K+/Na+ Homeostasis and Potassium Transport Capacity in the Boll-Leaf System under Moderate Salinity"

_agronomy, doi:10.3390/agronomy12122962_

Round 1
Reviewer 1 Report
Salinity of soils is a big and an ever increasing problem of agriculture. Looking for possibilities of salinity stress mitigation is an important issue therefore the undertaken research is justified and needed.
My comments:
1.Whether the issue „effects of K application on the uptake and distribution of K+ and Na+ in the cotton boll-leaf system under salt stress” is real important scientific problem and what practical significance has the solution of this problem?
2. In how many replications was conducted this experiment?
3. Did at April 15 were sowing seeds to soils with different salinity or with the same salinity?
4. Whether the dose 300 kg of K2O/ha is justified in the cotton production?
5. How the plants were watered in the containers?
6. Why the K+/Na+ ratio was presented as the mean from 2 years and the other results separately for the particular years of research? – it should be explained.
7. In the title of Table 6 should be add the unit of length (mm) because the data presented in the form : 2019 (mm) and 2020 (mm) are not correct.
8. In the Results is lack of an unequivocal information whether applied doses should be recommended to use in the cotton cultivation on the saline soils.
Author Response
Response to Reviewer 1 Comments
Point 1: Whether the issue "effects of K application on the uptake and distribution of K+ and Na+ in the cotton boll-leaf system under salt stress" is real important scientific problem and what practical significance has the solution of this problem?
Response 1: Cotton yield and fiber quality are difficult to improve due to the high salt content and low K+/Na+ ratio in saline-alkali soils. In the leaf-boll system, approximately 80% of the organic nutrients of the boll came from the leaf subtending the cotton boll (LSCB), and potassium application promoted the photosynthesis of the LSCB. However, there have only been a few reports to date on the relationships between salt stress and K in the alleviation effects of K application on the uptake and distribution of K+ and Na+ in the cotton boll-leaf system under salt stress. As a result, we investigated the physiological mechanism of K application in saline soils to regulate K+/Na+ homeostasis under salt stress. The regulatory effects of increasing K fertilizer application rates lay the theoretical foundation for increasing cotton fiber length under salt stress. There are specific articles to prove the real importance of the study of our laboratory.
[1] Ju, F. Y.; Pang, J. L.; Huo, Y. Y.; Zhu, J. J.; Yu, K.; Sun, L. Y.; Loca, D. A.; Hu, W.; Zhou, Z. G.; Wang, S. S.; Chen, B. L.; Tang, Q. X. Potassium application alleviates the negative effects of salt stress on cotton (Gossypium hirsutum L.) yield by improving the ionic homeostasis, photosynthetic capacity and carbohydrate metabolism of the leaf subtending the cotton boll. Field Crop Res. 2021, 272. https://doi.org/10.1016/J.FCR.2021.108288
Point 2: In how many replications was conducted this experiment?
Response 2: The experiments were conducted in 2019 and 2020, and we performed two replicates between the years. Each treatment had 3 replicates, each of which contained 20 cotton buckets.
Point 3: Did at April 15 were sowing seeds to soils with different salinity or with the same salinity?
Response 3: On April 15, seeds were sown into soils with the same low salinity, and then at the three-leaf stage, the seedling was transplanted into each bucket with different salinities.
Point 4: Whether the dose 300 kg of K2O/ha is justified in the cotton production?
Response 4: Cotton is a typical potassium-loving crop, with potassium being the second most important nutrient in the cotton plant after nitrogen. What is noteworthy is that cotton was transplanted, not directly seeded cotton in our research. The growth period of transplanting cotton is approximately 130 days, and the target seed cotton yield is more than 4500 kg/ha. In addition, our laboratory has conducted long-term research, and the results showed that 300 kg of K2O/ha is justified in cotton production. There are specific articles to prove it, and we have listed three of them.
[1] Hu, W., Yang, J.S., Meng, Y.L., Wang, Y.H., Chen, B.L., Zhao, W.Q., Zhou, Z.G. Potassium application affects carbohydrate metabolism in the leaf subtending the cotton (Gossypium hirsutum L.) boll and its relationship with boll biomass. Field Crops Research 2015, 179, 120-131. https://doi.org/10.1016/j.fcr.2015.04.017
[2] Yang, J.S., Hu, W., Zhao, W.Q., Chen, B.L., Wang, Y.H., Zhou, Z.G., & Meng, Y.L. Fruiting branch K+ level affects cotton fiber elongation through osmoregulation. Frontiers in Plant Science 2016, 7, 13. https://doi.org/10.3389/fpls.2016.00013
[3] Ju, F. Y.; Pang, J. L.; Huo, Y. Y.; Zhu, J. J.; Yu, K.; Sun, L. Y.; Loca, D. A.; Hu, W.; Zhou, Z. G.; Wang, S. S.; Chen, B. L.; Tang, Q. X. Potassium application alleviates the negative effects of salt stress on cotton (Gossypium hirsutum L.) yield by improving the ionic homeostasis, photosynthetic capacity and carbohydrate metabolism of the leaf subtending the cotton boll. Field Crop Res. 2021, 272. https://doi.org/10.1016/J.FCR.2021.108288
Point 5: How the plants were watered in the containers?
Response 5: Considering the problem of water uniformity in buckets, we used measuring cups for quantitative watering to ensure that the relative water content of each treated soil was approximately 75%. Water is poured evenly over the surface of the soil.
Point 6: Why the K+/Na+ ratio was presented as the mean from 2 years and the other results separately for the particular years of research? – it should be explained.
Response 6: To present the data concisely, we presented the average value in the past, but now we have supplemented the two-year data in the manuscript.
Point 7: In the title of Table 6 should be add the unit of length (mm) because the data presented in the form: 2019 (mm) and 2020 (mm) are not correct.
Response 7: Sorry, there is no Table 6 in our manuscript. We guess you want to mean Table 4. The unit of length (mm) has been added.
Point 8: In the Results is lack of an unequivocal information whether applied doses should be recommended to use in the cotton cultivation on the saline soils.
Response 8: In our results, K application alleviated the decrease in fiber length significantly, and K300 had a better alleviating effect. The results showed that K150 (i.e., K+/Na+=1/13) completely alleviated the reduction in fiber length of the salt-tolerant variety (CCRI 79) caused by S2 salt stress, while K300 (i.e., K+/Na+=1/10) entirely alleviated the reduction in fiber length of the salt-sensitive variety (Simian 3) caused by S2 salt stress.

Reviewer 2 Report
Overall this is an interesting job.
Τhe manuscript is clear, relevant for the field and presented in a well-structured manner.
The paper is written in good English.
My main concern is the suitability and feasibility of the experimental and analysis methodology. The statistical analysis as well as the experimental design are not clearly stated.
More comments on the subject can be found in the text.
Also, the extensive reference in the text of all the results presented in graphs, tires the reader and disorients from the perception of the important findings.

Author Response
Response to Reviewer 2 Comments
Dear Reviewers,
Thank you for the reviewers’ comments concerning our manuscript entitled “Potassium application increases cotton (Gossypium hirsutum L.) fiber length by improving K+/Na+ homeostasis and potassium transport capacity in the boll-leaf system under salt stress” (ID: agronomy-2021464). We have studied the comments carefully and have made corrections that we hope meet with approval.
First, we supplied the ambiguity in the experimental design. Each treatment had 3 replicates, each of which contained 20 cotton buckets using a randomized complete block design (RCBD) under a factorial arrangement.
Second, for the statistical analysis, the results were analyzed by ANOVA. The least significant difference (LSD) method was used to test the significance of the mean values at the P = 0.05 level. Variety and year were not included in the ANOVA. In terms of production, the salt-sensitive variety Simian 3 cannot be planted on the land of medium salt and high saline-alkali conditions. However, to clarify the regulation mechanism of the salt-tolerant variety CCRI 79 after potassium application, we used Simian 3 as the control, paying more attention to the regulating effect of potassium on salt.
Third, I am sorry for the extensive reference in the text of all the results presented in graphs, making you tired and disoriented from the perception of the important findings.
Our experiment was very complicated from the very beginning of the design. It consisted of 18 processes and required a large number of diagrams and tables to clarify it clearly. Part of the data must be listed to reflect our workload or our rigorous working attitude. All the data need to be listed to ensure the authenticity and reliability of the experiment. On this basis, we performed much work and came to the conclusion of Figure 6 after in-depth analysis. In addition, each paragraph of the text ends with a general conclusion to make it easier for reviewers to spot important conclusions.
As you suggested, the final conclusion is also divided into paragraphs, and conclusions should answer the questions and address the aims stated in the introduction.
In addition, we have revised and simplified the language expression of the manuscript to better present the important results of the study.
Finally, as suggested by the reviewer, some slipshod problems, such as typos and lack of clarity, have been corrected in the manuscript.
Thank you again for your positive comments and valuable suggestions to improve the quality of our manuscript.
Sincerely,
The Authors

Reviewer 3 Report
Zhu et al tried to show the Potassium function in homeostasis and potassium transport capacity in the boll leaf of cotton under salinity.
Cotton is a really important industrial crop that is worth investigating about its life under stress. also, I see material and the methods are chosen well.
However, I believe authors should respond to some questions:
1. references are too old... Why?
Salinity is a major problem, during the year hundreds of papers published in this area, and I encourage the team to use more recent references for salinity, potassium, and nitrogen, some recent papers suggested you may like to add.
https://www.sciencedirect.com/science/article/abs/pii/S0981942822000018
https://www.mdpi.com/2073-4395/8/3/31
https://www.nature.com/articles/s41598-022-14163-4
2. Some keywords are the same in the title and keywords.. why?
3. I encourage you not to use the plant name and the scientific name like this (cotton (Gossypium hirsutum L.)), just use the scientific name.
4. The manuscript should be rechecked by a native speaker.
Author Response
Response to Reviewer Comments
Point 1: References are too old... Why? Salinity is a major problem; during the year hundreds of papers published in this area, and I encourage the team to use more recent references for salinity, potassium, and nitrogen, some recent papers suggested you may like to add.
Response 1: We sincerely appreciate the valuable comments. We have checked the literature carefully and updated more references.
Point 2: Some keywords are the same in the title and keywords. why?
Response 2: Because these words are in line with the theme of our research, we have improved the keywords according to your suggestion.
Point 3. I encourage you not to use the plant name and the scientific name like this (cotton (Gossypium hirsutum L.)), just use the scientific name.
Response 3: Thank you for pointing this out. We marked it in the title and in the first appearance of the text, but used more plant names in the text for narrative convenience.
Point 4. The manuscript should be rechecked by a native speaker.
Response 4: Thank you for your suggestion. We have invited a friend who is a native English speaker to help polish our article. We hope the revised manuscript is acceptable.

Round 2
Reviewer 2 Report
Dear Editor,
authors presented a much improved version of the manuscript. They fixed almost all the issues that were pointed out to them, made necessary additions that clarified several ambiguities, and improved the writing language and the general picture for the reader.
However, I am afraid I will have to insist on the need to correct certain points.
1) in the introduction (lines 101 and 110) reference is made to the length of the fiber and to cotton yield, which is not the subject of the study. Authors should only refer to the parameters under study.
2) In methods and materials, although authors clarified the experimental design enough and gave details about irrigation after the reviewers' comments, there are no relevant references in the text. The irrigation method, frequency and quantities should be added in detail so that the experiment can be repeated.
3) On line 168 authors must correct the heteroreference to the correct one Gipson and Ray (1969).
4) In the diagrams of Figures 2, 3, 4, 5 the scale of the y axis should be the same everywhere in order to facilitate comparisons by the readers.
5) I do not understand the reason for the existence of image 6, which I also consider misleading in relation to the findings. The footnote is also misleading because it does not describe a model in the image but management.
6) The conclusions, although improved and now following the questions posed in the introduction, still need correction. Authors should highlight the importance of their research, the contribution to addressing the problem of fiber length reduction in saline soils and propose further investigation of the subject.
Author Response
Point 1: in the introduction (lines 101 and 110) reference is made to the length of the fiber and to cotton yield, which is not the subject of the study. Authors should only refer to the parameters under study.
Response: Thank you. Fiber length is included in our study, so some are listed, and some expressions that are not relevant to this study have been deleted.
Point 2: In methods and materials, although authors clarified the experimental design enough and gave details about irrigation after the reviewers' comments, there are no relevant references in the text. The irrigation method, frequency and quantities should be added in detail so that the experiment can be repeated.
Response: Thank you. Considering the problem of water uniformity in buckets, we used measuring cups for quantitative watering to ensure that the relative water content of each treated soil was the same. Each bucket was irrigated to maintain a 75± 5% relative soil water content (RSWC). The soil moisture level of all buckets was maintained by following the method adapted by Liu et al. (2008). There are specific articles that can be referenced, and we have listed two of them.
[1] Liu, R. X.; Zhou, Z. G.; Guo, W. Q.; Chen, B. L.; Osterbuis, D. M. Effects of N fertilization on root development and activity of water-stressed cotton (Gossypium hirsutum L.) plants. Agric. Water Manage 2008, 95, 1261–1270. https://doi.org/10.1016/j.agwat.2008.05.002
[2] Rizwan, Z.; Haoran, D.; Muhammad, A.; Zhao, W.Q.; Wang, Y. H.; Zhou, Z. G. Potassium fertilizer improves drought stress alleviation potential in cotton by enhancing photosynthesis and carbohydrate metabolism. Environmental and Experimental Botany 2017, 137, 73-83. https://doi.org/10.1016/j.envexpbot.2017.02.002
Point 3: On line 168 authors must correct the heteroreference to the correct one Gipson and Ray (1969).
Response: Thank you. Revised.
Point 4: In the diagrams of Figures 2, 3, 4, 5 the scale of the y axis should be the same everywhere in order to facilitate comparisons by the readers.
Response: Thank you. The scale of the y axis in the diagrams of Figs. 2, 3, 4, and 5 has been modified.
Point 5: I do not understand the reason for the existence of image 6, which I also consider misleading in relation to the findings. The footnote is also misleading because it does not describe a model in the image but management.
Response: Thank you. We also find that Image 6 is less closely linked to the article. Image 6 has been deleted.
Point 6: The conclusions, although improved and now following the questions posed in the introduction, still need correction. Authors should highlight the importance of their research, the contribution to addressing the problem of fiber length reduction in saline soils and propose further investigation of the subject.
Response: Thank you. Revised. In our results, K application alleviated the decrease in fiber length significantly, and K300 had a better alleviating effect. The results showed that K150 (i.e., K+/Na+=1/13) completely alleviated the reduction in fiber length of the salt-tolerant variety (CCRI 79) caused by S2 salt stress, while K300 (i.e., K+/Na+=1/10) entirely alleviated the reduction in fiber length of the salt-sensitive variety (Simian 3) caused by S2 salt stress. As a result, the regulatory effects of increasing K fertilizer application rates on K+/Na+ homeostasis and potassium transport capacity in the boll-leaf system under salt stress conditions are revealed, laying the theoretical foundation for increasing cotton fiber length under salt stress.
In plants, the earliest response to salt stress and K fertilizer occurrs in the root system. In the future, based on our existing studies, the absorption, transport, and distribution characteristics of K+ and Na+ in cotton root of different salt-tolerant cotton cultivars will be further discussed.

Reviewer 3 Report
In general, it looks acceptable. However, I still believe salt stress manuscripts should cover with more recent references.
Author Response
Point: In general, it looks acceptable. However, I still believe salt stress manuscripts should cover with more recent references.
Response: Thank you. Revised.
